# Interaction of the Exopolysaccharide from *Lactobacillus plantarum* YW11 with Casein and Bioactivities of the Polymer Complex

**DOI:** 10.3390/foods10061153

**Published:** 2021-05-21

**Authors:** Min Zhang, Tiantian Lai, Mengke Yao, Man Zhang, Zhennai Yang

**Affiliations:** Beijing Advanced Innovation Center for Food Nutrition and Human Health, Beijing Engineering and Technology Research Center of Food Additives, Beijing Technology and Business University, Beijing 100048, China; 1850201008@st.btbu.edu.cn (M.Z.); 1930201070@st.btbu.edu.cn (T.L.); 1930201080@st.btbu.edu.cn (M.Y.); 1930202084@st.btbu.edu.cn (M.Z.)

**Keywords:** *Lactobacillus plantarum* YW11, exopolysaccharide, casein, interaction, bioactivities

## Abstract

There has been an increased application of exopolysaccharide (EPS)-producing lactic acid bacteria (LAB) in fermented dairy products, but interactions between EPS and casein (CAS), and bioactivities of their complex are poorly studied. In this study, EPS produced by *Lactobacillus plantarum* YW11 (EPS-YW11) was studied for interactions with CAS in a simulated fermentation system acidified by D-(+)-gluconic acid δ-lactone. The results showed that there was interaction between EPS-YW11 and CAS when EPS (up to 1%, *w*/*v*) was added to the casein solution (3%, *w*/*v*) as observed with increased viscoelasticity, water holding capacity, ζ-potential and particle size of EPS-YW11/CAS complex compared with CAS alone. Microstructural analysis showed that a higher concentration of EPS facilitated more even distribution of CAS particles that were connected through the polysaccharide chains. Infrared spectroscopy further confirmed interactions between EPS and CAS by intermolecular hydrogen bonding, electrostatic and hydrophobic contacts. Further evaluation of the bioactivities of EPS-YW11/CAS complex revealed significantly increased antibiofilm, antioxidation, and bile acids binding capacity. The present study provides further understanding on the mechanism of interactions between EPS produced by LAB and CAS, which would benefit potential applications of EPS in fermented dairy products with enhanced functionality.

## 1. Introduction

The exopolysaccharide (EPS) produced by lactic acid bacteria (LAB) is a type of high molecular weight carbohydrate polymer secreted outside the cell wall during their growth [1]. EPS has been reported with various bioactivities, such as antioxidation, anti-tumor, anti-bacteria, anti-mutagen, immune regulation, and adjustment of gastrointestinal floral balance [2]. During the last decade, EPS-producing LAB have been increasingly used in fermented dairy foods in the improvement of texture and mouthfeel of the products [3]. It has been reported that EPS can interact with milk proteins to generate spatial barriers among themselves, thus changing the viscosity, rheology and taste of fermented dairy products, preventing whey separation and gel breakage in yogurt [4,5]. Interaction between EPS and proteins also affected aggregation properties of milk protein clusters due to the formation of a network structure [6], but the mechanism of interactions between these polymers is not well understood.

Casein (CAS), which consists of α_s1_-, α_s2_-, β-, and κ-casein at a weight ratio of 4:1:4:1, constitutes the major part of milk proteins [7]. CAS is present as stable micellar structures in aqueous solutions due to the molecular self-assembly of its distinct hydrophobic and hydrophilic domains [8]. The amphiphilicity of CAS molecules enhances their surface activity, water-holding capacities, and emulsifying properties, as well as the ability to bind to ions or small molecules [9]. CAS has been increasingly used to improve the texture, shelf life, and nutritive value of emulsions by interaction with biomolecules to form complexes and conjugates with synergistic combinations of properties [10]. The interaction between CAS and EPS was dependent on the system environment (pH, temperature, ionic strength, etc.), and the structural properties of these two substances (molecular size, charge, and charge distribution, etc.) [10]. Fermented milk containing EPS that might interact with CAS was found with greater shear stress, hysteresis loop area, viscosity, water-holding capacity, and gel strength [11]. EPS was found to promote agglomeration of CAS micelles and recovery of gel structure of yogurt upon shearing [12].

*Lactobacillus plantarum (L. plantarum)* YW11 was previously isolated from Tibet kefir in our laboratory, and it was shown to produce an EPS (EPS-YW11) at 131.26 mg/L in a semi-defined medium (SDM) [13]. EPS-YW11 was characterized as an anionic polysaccharide composed of glucose and galactose at a molar ratio of 2.71:1, with the possible presence of N-acetylated sugar residues. EPS-YW11 exhibited a relatively strong ability of binding moisture and high heat stability due to its highly branched and porous structure [14,15]. EPS-YW11 also possessed several bioactivities, e.g., anti-tumor, anti-ulcer, serum cholesterol reduction and immunoregulation [16]. In the present study, interaction between EPS-YW11 and CAS was studied under acidic conditions by using glucoronic acid (GDL) as an acidifying agent to simulate the lactic fermentation process [17]. Formation of the EPS-YW11/CAS complex under the acidic condition was evaluated by various analyses including microrheology, texture profile analysis (TPA), particle size distribution, transmission electron microscopy (TEM), ζ-potential, and infrared spectroscopy (IR). Bioactivities of the EPS-YW11/CAS complex such as bile acid binding, antioxidant activities and antibiofilm were also studied. This study provides further understanding of the mechanism of interactions between EPS-YW11 and CAS, which may facilitate exploitation of the potential application of EPS-producing LAB in functional fermented dairy products.

## 2. Materials and Methods

### 2.1. Materials

*L. plantarum* YW11 was maintained as frozen (−80 °C) stocks in SDM medium supplemented with 20% (*v*/*v*) glycerol and adjusted to pH 6.6 with 1 M acetic acid [18,19]. *L. plantarum* YW11 was revitalized in SDM medium by two overnight sub-culturing events at 37 °C before use for preparation of EPS.

CAS was purchased from Beijing Mreda Technology Co. Ltd. (Beijing, China), and D-(+)-gluconic acid δ-lactone (GDL) from Beijing Hua Wei Rui Ke Chemical Co. Ltd. (Beijing, China), chenodeoxycholic acid (CDCA) from Sigma–Aldrich (St. Louis, MO, USA), total bile acids (TBA) kit and total antioxidant capacity (T-AOC) kit from Nanjing Jiancheng Bioengineering Institute. Potassium dihydrogen phosphate, potassium citrate, potassium sulfate and calcium chloride were purchased from Beijing Pinellia Technology Development Co., Ltd. (Beijing, China). Sodium hydroxide, sodium citrate, and potassium chloride were purchased from Shanghai Aladdin Biochemical Technology Co., Ltd. (Shanghai, China). Sodium dihydrogen phosphate and disodium hydrogen phosphate were purchased from Sinopharm Group Chemical Reagent Co. Ltd. (Beijing, China). All reagents were of analytical grade.

### 2.2. Preparation of EPS Samples

EPS produced by *L. plantarum* YW11 in a SDM was isolated by ethanol precipitation, purified by DEAE-cellulose and sepharose CL-6B chromatography, and then analyzed as described earlier [14]. The purified EPS sample, which contained 92.35 ± 2.38% polysaccharide, 2.52 ± 0.12% moisture, 1.56 ± 0.09% uronic acids and 1.38 ± 0.25% protein as previously determined [14], was used in the preparation of EPS-YW11/CAS sample for our study.

### 2.3. Preparation of EPS-YW11/CAS Samples

Simulated milk ultrafiltrate (SMUF) was prepared using 1.58 g/L potassium dihydrogen phosphate, 1.2 g/L potassium citrate, 2.12 g/L sodium citrate, 0.18 g/L potassium sulfate, 1.32 g/L calcium chloride, 0.65 g/L magnesium chloride, 0.30 g/L potassium carbonate, and 0.60 g/L potassium chloride [20]. The SMUF was adjusted to pH 6.7 with 0.1 M sodium hydroxide. EPS-YW11/CAS complex samples for different analyses of this study were prepared as described previously [21,22]. Briefly, CAS powder (3%, *w*/*v*) was added to SMUF, stirred for 20 h at 4 °C, then stirred for 30 min at 95 °C till it was completely dissolved. Subsequently, different proportions (0%, 0.25%, 0.50%, 0.75%, and 1%) of EPS were added to the CAS suspension in SMUF (20 mL), and the pH was adjusted to 6.7 after EPS was completely dissolved. GDL (1%) was then added to the mixture to be maintained at 42 °C, and the change of pH and rheology was monitored till the pH reached 4.5. The iCinac dairy fermentation monitor (AMS Alliance, Rome, Italy) was used to monitor the pH changes during the simulated fermentation process. Data were collected every 1 min and measured for 2.5 h.

### 2.4. Microrheological Analysis

Microrheological analysis was performed during the simulated fermentation process of the samples upon addition with GDL. The sample solution (20 mL) was transferred into a special sample cell of the microrheometer (Rheolaser Master, Formulation Inc., Toulouse, France), and the test program was run. The changes in the elasticity index (EI), viscosity index (MVI), and fluidity index (FI) of the samples were monitored over time during the acidification process at 42 °C. Data were collected every 2 min and measured for 2 h.

### 2.5. Texture Profile Analysis

Analysis of hardness, viscosity, cohesion, and adhesion of the samples was performed with a texture analyzer (Brook-Field, Middleboro, MA, USA), using a cylindrical TA10 probe at 10 mm test distance, at 2.0 mm/s pre-test speed, 0.50 mm/s test speed, 0.50 mm/s return speed, and 20.0 points/s data frequency over three loop tests.

### 2.6. Water-Holding Capacity Analysis

Water-holding capacity (WHC) of the CAS clot sample was determined in a centrifuge tube. Centrifugation was performed at 25 °C at 2800× *g* for 10 min. The supernatant was removed, and the centrifuge tube was inverted for 10 min and then immediately weighed [23].
WHC (%) = weight of centrifuge sediment/weight of sample × 100%(1)

### 2.7. Measurement of ζ-Potential

The electrical characteristics (ζ-potential) of the particles in solution were determined by measuring the electrophoretic mobility (Zetasizer Nano-ZS90, Malvern Panalytical, Malvern, UK). Sample solutions (2.0 mg/mL) were in the folded capillary cell and sealed with two stoppers. The cells were then mounted to determine the ζ-potential of the molecules. The ζ-potential tests were performed in triplicate.

### 2.8. Particle Size Analysis

The particle size distribution of the samples was determined with a laser particle size analyzer (LS13320, Beckman Coulter, CA, USA). After the sample was stirred evenly, it was slowly added dropwise to the water tank of the particle size analyzer till the test system reached the turbidity of 40%, then the particle size was determined. All samples were measured three times.

### 2.9. Infrared Spectral Analysis

The infrared spectra of the samples were acquired by using the total-reflectance mode of a Fourier transform infrared (FT-IR) spectrometer (Nicolet 6700, Thermo Fisher, Waltham, MA, USA). Samples were freeze-dried and mixed with potassium bromide (KBr) at a ratio of 1:100, and then compressed into tablets. Measurements were performed in the mid-infrared region (400–4000 cm^−1^). All samples were analyzed three times under the same conditions.

### 2.10. Transmission Electron Microscopy

TEM analysis was performed to observe the molecular morphology of the sample. To prepare TEM samples, one drop of the complex suspension was placed on a copper grid and stained with 2% phosphotungstic acid, which was then air dried overnight. The grid was placed in the microscope for imaging at 100 kV accelerating voltage, while the images were taken on a Gatan electron energy loss spectrometry system using a 6 eV energy slit [24].

### 2.11. Bile Acid Binding Capacity

The bile acid binding capacity of the samples was examined using a reported method slightly modified [25]. In brief, each EPS-YW11/CAS sample (10 mg) was added to 1 mL of 1 mM CaCl_2_ solution. Subsequently, 100 μL of 0.01 M HCl was added while gently shaking at 37 °C. 10 µL of 0.01 M NaOH was added to neutralize the mixture, and then 500 μL of 400 mM CDCA solution was added. Reaction in the simulated intestinal fluid was carried out at 37 °C for 1 h. A total of 100 µL of supernatant was taken, and a TBA kit was used to determine bile acid. The absorbance was measured at 405 nm.

### 2.12. Determination of Total Antioxidant Capacity

The EPS-YW11/CAS sample was diluted four times to obtain a transparent solution. According to the kit instructions of the Nanjing Jiancheng Institute of Biological Engineering, T-AOC of the sample was measured in a 96-well microplate. The OD value was measured at 405 nm using a microplate reader (Thermo, USA). T-AOC of the sample was calculated as follows: T-AOC = Asample − Acontrol. Asample is the absorbance of the sample and ABTS (2,2′-azino-bis 3-ethylbenzthiazoline-6-sulfonic acid) working solution, and Acontrol is the absorbance of water and ABTS working solution [2].

### 2.13. Assay of In Vitro Anti-Biofilm Activity

The concentration of activated *Staphylococcus aureus* CMCC 26071, *Enterobacter sakazakii* CICC 21544, *Shigella*
*flexneri* CICC 21534, and *Salmonella typhimurium* CICC 22,956 was adjusted to 1.0 × 10^8^ cfu/mL. The EPS-YW11/CAS sample was formulated as an aqueous solution with a concentration of 1.0 mg/mL (0.22 μm membrane filtration sterilization). A total of 100 µL of bacterial solution was taken, and 100 μL of the sample was mixed with it evenly and added to a 96-well microplate. The negative control well contained only medium. The samples were incubated at 37 °C for 24 h. Then the wells were washed, and the adhered cells stained with 2% (*w*/*v*) crystal violetto, which was solubilized with 0.16 mL of 33% (*v*/*v*) glacial acetic acid per well for the measurement of optical density (OD) at 590 nm. The biofilm inhibition rate was calculated as follows: inhibition rate (%) = [1 − (OD_sample_/OD_control_)] × 100%. Each data point was averaged from three replicate wells, and the standard deviation (SD) was calculated.

### 2.14. Statistical Analysis

All experiments were performed in triplicate, and the experimental data were analyzed by one-way ANOVA with Fisher’s least significant difference (LSD) method by using SPSS 22.0 software (IBM, Armonk, NY, USA). *p* < 0.05 indicated significant difference. Origin 8.5 software (OriginLab, Northampton, MA, USA) was used for mapping.

## 3. Results and Discussion

### 3.1. pH Analysis of GDL-Simulated Fermentation Process

GDL can be hydrolyzed to produce gluconic acid, which lowers the pH of the milk system and causes protein to aggregate, precipitate and form gels [26,27]. Using GDL to simulate the fermentation process of yogurt has the advantage of good repeatability and can avoid potential differences caused by the growth of lactic acid bacteria [12]. The pH change of the EPS-YW11/CAS complex during the GDL-simulated fermentation process over time is shown in Figure 1A. The pH curves of all samples had roughly the same decreasing trend of pH with the acidification process by GDL. An increasing concentration of EPS resulted in a faster decrease of pH in the EPS-YW11/CAS complex, while CAS alone showed the slowest decrease of pH. This suggested that the interaction between EPS-YW11 and CAS was beneficial to the formation of the clot upon decreasing pH [28].

### 3.2. Microrheological Analysis

Microrheological analysis can be performed with an optical microrheometer to monitor the movement of particles without damaging the sample [29]. During the fermentation process of yogurt, the microstructure and rheological properties of the milk-based mixture upon inoculation with the LAB culture change significantly from fluid to formation of a gel structure [30]. Plots of elastic index (EI), macroscopic viscosity index (MVI), and fluidity index (FI) as a function of time reflect changes of the elasticity, viscosity, and fluidity of the sample, respectively [31]. The rheological change of the GDL-simulated fermentation process over time of the EPS-YW11/CAS complex sample is shown in Figure 1B–D. The EI, MVI, and FI of all samples had roughly the same trend over time, but the specific values of the rheological properties under different conditions showed certain differences. The viscoelastic changes of each group of samples in the acidification process with time could be roughly divided into three stages. In the early stage of acidification, the samples were in a state of low viscosity, low elasticity, and high fluidity. Subsequently, within 20–60 min of acidification, there were rapid changes in elasticity, viscosity and fluidity of the samples due to protein aggregation and gelation reaction (Figure 1A). Specifically, the elasticity increased significantly (*p* < 0.05) when the addition of EPS-YW11 was increased to 1%, indicating that the interaction between EPS and CAS improved viscoelasticity of the EPS-YW11/CAS complex. Previously, excluding the EPS effect, GDL induced gelation of milk causing an obvious decrease of viscoelasticity and fluidity when the pH value of milk decreased to about 5.2. In the final stage, with further extension of time, the changes in elasticity, viscosity, and fluidity of the EPS-YW11/CAS complex slowed down when the pH reached 4.5, and a stable gel formed with a slight increase in MVI and decrease in FI probably due to a certain degree of dehydration of the samples [32].

### 3.3. Texture Profile Analysis

A texture profile analysis by simulating human chewing of food has been widely used to evaluate textural properties of solid and semi-solid foods [33]. The texture of fermented food is closely related to its internal structure as reflected by its hardness, viscosity, and adhesiveness [34]. As shown in Table 1, increasing the addition of EPS-YW11 from 0.25% to 1% (*w*/*v*) resulted in decreased hardness, adhesiveness and increased viscosity of the EPS-YW11/CAS complex samples, but cohesion of the samples were not obviously affected (*p* > 0.05). This might be due to increased interactions of EPS with CAS when more EPS was added, leading to less hard and adhesive samples [35,36,37]. Previously, Zhang et al. [38] reported that EPS could affect the textural properties of a yogurt clot by decreasing its hardness.

### 3.4. Water-Holding Capacity

WHC is an indicator of the ability of a gel to bind water. The better the WHC of the gel, the stronger the force of the gel to bind water, and the better the stability of the gel [39]. The WHC of fermented milk played a key role in extending the shelf life of the products [40]. Figure 2A shows that the WHC of the EPS-YW11/CAS samples increased significantly (*p* < 0.05) when more EPS was added. The WHC reached 56.7% when EPS was at 1% (*w*/*v*), which was 19.9% higher than that obtained with CAS alone. Previously, EPS was shown to significantly increase the viscosity of the clot and reduce dehydration [41]. Dehydration of the clot system during the storage of yogurt might cause weakening of the gel network structure [42]. In the presence of EPS, the gel did not undergo dehydration immediately under stirring conditions [33]. Interaction of EPS with bound water, CAS, and colloidal particles played an important role in improving WHC and limiting syneresis of dairy products [43]. The milk fermented by the strain Ldb2214 producing EPS showed good WHC value [43].

### 3.5. ζ-Potential Analysis

ζ-Potential is typically used to describe the surface charge of colloidal particles in colloidal chemistry. The greater the absolute value of the ζ-potential, the more stable the colloid [44]. The results in Figure 2B showed that the potential of the samples was always negative. The absolute value of the ζ-potential increased with the addition of EPS-YW11, reaching the maximal value of −13.76 mV at the EPS concentration of 0.75% (*w*/*v*), which suggested the presence of electrostatic interactions between EPS and CAS [45]. In the whey protein concentrate solution added with pullulan, the electrostatic interactions also played a great role during the gelation [46]. The enhanced charge density of molecules or dispersed particles upon addition with EPS suggested improved stability of the mixture [47]. The addition of EPS333 to the CAS solution at pH 6.0 was found to increase ζ-potential and thus improve stability of the mixture [48].

However, when the addition of EPS increased to 1% (*w*/*v*), the absolute value of the ζ-potential decreased, which might be due to the steric barrier caused by excess EPS molecules, leading to a weakened electrostatic interaction between the particles in the system and thus reduced stability of the system [49]. Therefore, a change in the ζ-potential of the system, as described above, indicated a possible shift in the affinity type between CAS and EPS-YW11 during acidification.

### 3.6. Particle Size Analysis

The interaction of biopolymers may result in changes of the particle size and distribution of the dispersion system that in turn affect its surface area, turbidity, bulk density, and macroscopic properties [50]. As shown in Figure 2C, the increasing addition of EPS-YW11 up to 1% (*w*/*v*) resulted in increased particle size of the EPS-YW11/CAS complex (870–1150 nm). This might be due to the increased interaction between EPS and CAS molecules to form more macromolecular complexes as the strength of the force between the protein molecules reduced [38]. Electrostatic interaction was found to be the major driving force for polysaccharide/CAS complexation, and its particle size was dependent on the concentration of the polysaccharide [24].

### 3.7. Infrared Spectral Analysis

FT-IR spectroscopy is an effective method to evaluate the binding interaction for complex formation and the secondary structure changes in CAS after the addition of polysaccharide [24]. As shown in Figure 2D, the FT-IR spectrum of the EPS-YW11/CAS complex samples demonstrates a broad peak at 3396 cm^−1^, which is attributed to the strong –OH stretching vibration absorption [51]. The strong absorption peak at 1653 cm^−1^ corresponds to the stretching vibration of the C=O bond, and the absorption at 1536 cm^−1^ to the C-N bending of amides II of protein [14]. The absorption peak around 1232 cm^−1^ is attributed to the N-H bending of amides III of protein. The absorption peak around 1056 cm^−1^ and 580 cm^−1^ might be a symmetrical absorption peak of the pyranose ring C–O–C [52]. The intensity change and spectral shift at 1653 cm^−1^ (mainly C=O stretch) are related to the change of the secondary structure of the protein [53,54].

Figure 2D also shows that the absorption peaks of CAS (without addition of EPS) do not shift significantly after binding with EPS-YW11 at the concentration from 0 to 1% (*w*/*v*). The increased intensities of the absorption peaks of CAS suggest increased random coil structures of the protein upon addition of more EPS. The change in the intensity of the absorption peak (at 1653 cm^−1^) is due to EPS-YW11 binding to the C=O of CAS via hydrogen bonding and hydrophobic contacts [55]. Moreover, the broadening of the absorption peak at 3396 cm^−1^ for CAS also verifies the existence of hydrogen bonds and O–H stretching in EPS-YW11/CAS complexes, indicating that the intermolecular hydrogen bonds contribute to interactions of polymers [56]. Therefore, FT-IR spectroscopy confirmed the presence of molecular interactions between CAS and EPS-YW11 in different forms of force, and the secondary structure of CAS changed due to its binding with the polysaccharide.

### 3.8. Microstructural Characteristics

Microstructures of the EPS-YW11/CAS complex samples as observed by TEM in comparison with those of EPS-YW11 and CAS are shown in Figure 3. EPS-YW11 was shown to be a highly branched and porous structure (Figure 3A), while CAS was randomly distributed in a spherical shape (Figure 3B). After acidification with GDL, CAS aggregates formed (Figure 3C). However, with the increasing addition of EPS-YW11 up to 1% (*w*/*v*), the CAS aggregates were dispersed to smaller aggregates (Figure 3D,E), and they became more evenly distributed with CAS connected and distributed in the composite through the polysaccharide chains (Figure 3F,G) [57]. Similarly, the CAS/κ-carrageenan complex existed in aggregates with CAS micelles connected to each other by κ-carrageenan chains and randomly distributed in the aggregates as revealed by TEM [58]. Thus, interactions between CAS and EPS-YW11 resulted in more uniform texture of CAS/EPS-YW11 complex that was beneficial for application in foods with improved textural properties.

### 3.9. Anti-Biofilm Activities of EPS-YW11/CAS Complex

Some EPSs produced by LAB may act as signaling molecules to regulate gene expression involved in bacterial biofilm formation, thus mediating anti-biofilm activity [59]. EPS (5.0 mg/mL) from *L. plantarum* YW32 was shown with a strong ability to inhibit biofilm formation by several pathogens such as *Shigella flexneri* (44.67%), *Staphylococcus aureus* (45.13%), and *Salmonella typhimurium* (44.04%) [60].

As shown in Figure 4, EPS produced by *L. plantarum* YW11 showed different anti-biofilm activities against *Salmonella typhimurium* CICC 22956, *Enterobacter sakazakii* CICC 21544, *Staphylococcus aureus* CMCC 26071, and *Shigella flexneri* CICC 21534. CAS alone had no effect on the formation of biofilm (data not shown). However, all EPS-YW11/CAS complex samples with different concentration of EPS from 0.25 to 1% (*w*/*v*) exhibited significantly higher (*p* < 0.05) anti-biofilm activities than EPS alone. Increased activities of EPS-YW11/CAS complex samples were observed with higher concentrations of EPS, reaching the highest activity (60.21%) on *Staphylococcus*
*aureus* CMCC 26,071 at 1% of EPS. This indicated the existence of interactions between EPS-YW11 and CAS that played an important role in the enhanced anti-biofilm activity of the EPS-YW11/CAS complex.

### 3.10. Antioxidant Activity of EPS-YW11/CAS Complex

The antioxidant activity of samples can be evaluated by determining their total antioxidant capacity. As shown in Figure 5A, the antioxidant activities of all EPS-YW11/CAS complex samples were higher than those of EPS-YW11 and CAS alone. The antioxidant activity of the EPS-YW11/CAS complex increased with the increase in EPS concentration, reaching the highest value (0.72) at 1% (*w*/*v*) of EPS. This indicated interaction between EPS and CAS that increased antioxidant activity of the EPS-YW11/CAS complex. Previously, EPS-YW11 (up to 3.0 mg/mL EPS) was shown with strong antioxidant activity by both in vivo and in vitro tests [16]. The antioxidant effect of polysaccharides was related to their structural characteristics such as chain conformation, monosaccharide composition, and molecular weight of the polymers [61]. Therefore, interactions between EPS and CAS resulted in increased antioxidant capacity of the EPS-YW11/CAS complex probably due to the change of the polysaccharide structure.

### 3.11. Bile Acid Binding Capacity of EPS-YW11/CAS Complex

Macromolecular networks with tighter connections and more branches were shown to possess a better ability of binding bile acids in relation to the concentration [62]. Polysaccharides could effectively bind bile acid, exhibiting good hypolipidemic effects in vivo [63]. EPS produced by *L. plantarum* YW11, which had a highly branched and porous structure [14], was capable of binding bile acids in a concentration dependent manner (Figure 5B). Although CAS alone could not bind bile acids (data not shown), the EPS-YW11/CAS complex showed a higher capacity of binding bile acids than EPS alone. The bile acid binding capacity of the EPS-YW11/CAS complex increased with increase in EPS concentration from 0.25 to 1% (*w*/*v*), reaching the highest value (0.244 mg/g) at 1% (*w*/*v*) of EPS. This indicated interactions between EPS and CAS that resulted in the formation of a more complex network structure, promoting the binding of bile acids by the EPS-YW11/CAS complex. Previously, polysaccharides extracted from loquat leaves exerted remarkable in vitro binding capacities for bile acids [64]. Moringa oleifera leaf polysaccharide fraction with the highest galactose content and a large proportion of linear macromolecules exhibited the strongest bile acid binding capacity with potential hypolipidemic effects [65].

## 4. Conclusions

In this study, interactions between EPS (0%, 0.25%, 0.5%, 0.75%, and 1%, *w*/*v*) produced by *L. plantarum* YW11 and CAS were studied by different analytical methods, and bioactivities of EPS-YW11/CAS complex were evaluated. During the GDL-simulated acidification process, the increased addition of EPS caused a faster decrease of pH of the complex. Microrheological studies showed that the viscoelasticity of the EPS-YW11/CAS complex was improved when compared with CAS alone. Addition of EPS at higher concentrations increased WHC and viscosity, but decreased hardness and adhesiveness of the EPS-YW11/CAS complex, though cohesion of the samples was not obviously affected (*p* > 0.05).

Complexation between EPS-YW11 and CAS also increased the particle size and ζ-potential of the complex. Infrared spectroscopy confirmed interactions between EPS and CAS by intermolecular hydrogen bonding, and electrostatic and hydrophobic contacts, which caused changes of the secondary structure of CAS. Further microstructural analysis showed a more uniform texture of CAS/EPS-YW11 complex could be formed with increased EPS concentration, which was beneficial for application in foods with improved textural properties. Evaluation of the bioactivities of the EPS-YW11/CAS complex revealed significantly improved antibiofilm, antioxidation, and bile acids binding capacity compared with those of EPS-YW11 or CAS alone. Therefore, it would be of interest to further explore EPSs produced by LAB by employing effective interactions with casein for potential applications in functional dairy products.

## Figures and Tables

**Figure 1 foods-10-01153-f001:**
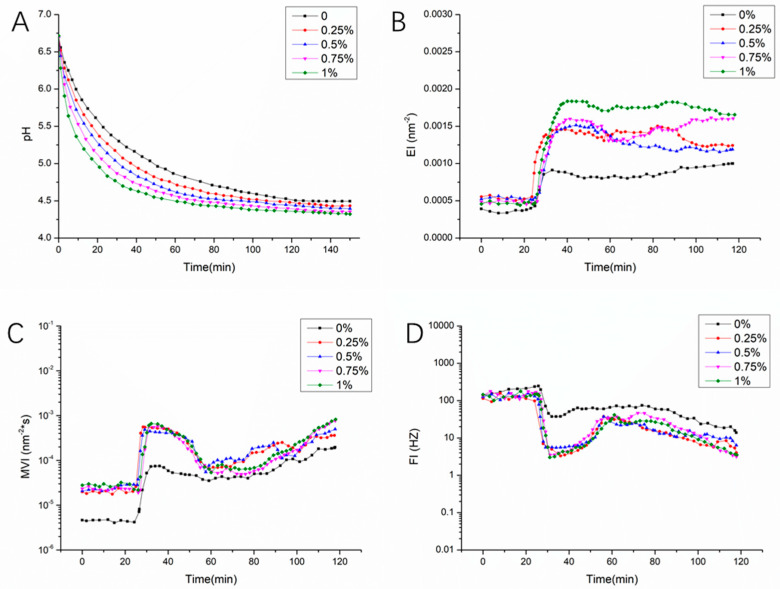
The pH and microrheological analyses of EPS−YW11/CAS complex during the GDL−simulated fermentation process. (**A**): change in pH; (**B**): change in EI; (**C**): change in MVI; (**D**): change in FI of EPS−YW11/CAS complex with increasing EPS concentration up to 1% (*w*/*v*). EPS−YW11: exopolysaccharide produced by *L. plantarum* YW11; CAS: casein; GDL: D− (+)−gluconic acid δ-lactone; EI: elastic index, MVI: macroscopic viscosity index; FI: fluidity index.

**Figure 2 foods-10-01153-f002:**
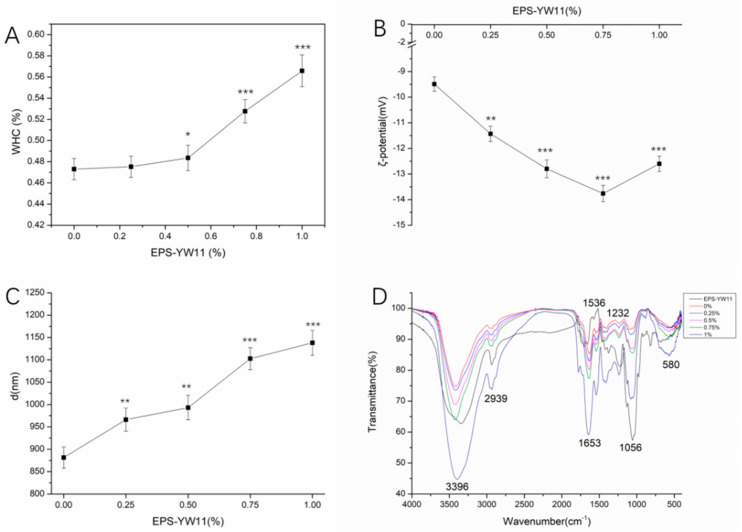
Changes of water-holding capacity, ζ-potential, particle size and infrared spectra of EPS−YW11/CAS complex during the GDL−simulated fermentation process. (**A**): change of water−holding capacity; (**B**): change of ζ−potential; (**C**): change of particle size; (**D**): change of infrared spectra of EPS−YW11/CAS complex with increasing EPS concentrations up to 1% (*w*/*v*). The results are represented as mean ± SD (*n* = 3). Values with different stars were significantly different from the control group (* *p* < 0.05, ** *p* < 0.01, *** *p* < 0.001). EPS−YW11: exopolysaccharide produced by *L. plantarum* YW11; CAS: casein; GDL: D−(+)−gluconic acid δ-lactone.

**Figure 3 foods-10-01153-f003:**
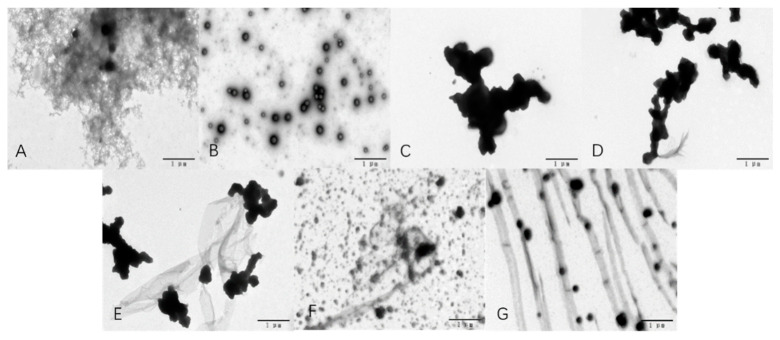
TEM images of EPS-YW11, CAS and EPS-YW11/CAS complex with increasing EPS concentrations up to 1% (*w*/*v*). (**A**): image of EPS-YW11; (**B**): image of CAS without acidification; (**C**): image of acidified CAS; (**D**): image of EPS−YW11/CAS complex at 0.25% EPS−YW11; (**E**): image of EPS−YW11/CAS complex at 0.5% EPS−YW11; (**F**): image of EPS−YW11/CAS complex at 0.75% EPS−YW11; (**G**): image of EPS−YW11/CAS complex at 1% EPS−YW11. EPS−YW11: exopolysaccharide produced by *L. plantarum* YW11; CAS: casein.

**Figure 4 foods-10-01153-f004:**
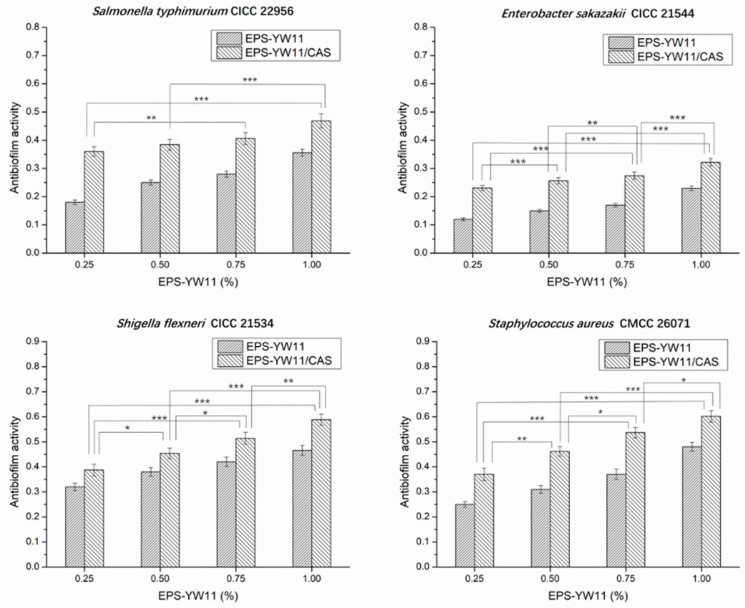
Anti−biofilm activity of EPS−YW11 and EPS−YW11/CAS complex against *Salmonella*
*typhimurium* CICC 22956, *Staphylococcus aureus* CMCC26071, *Shigella flexneri* CICC 21534, and *Enterobacter sakazakii* CICC21544. The results are represented as mean ± SD (*n* = 3). Values between each pair of columns connected with a line marked with different stars on the top were significantly different (* *p* < 0.05, ** *p* < 0.01, *** *p* < 0.001). EPS-YW11: exopolysaccharide produced by *L. plantarum* YW11; CAS: casein.

**Figure 5 foods-10-01153-f005:**
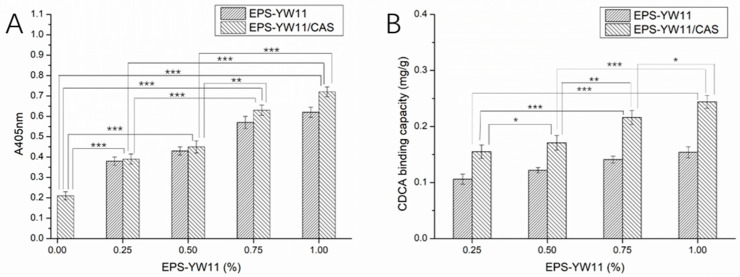
Antioxidant activity (**A**) and bile acid binding capacity (**B**) of EPS−YW11 and EPS-YW11/CAS complex with increasing EPS concentration from 0.25% to 1% (*w*/*v*). The results are represented as mean ± SD (*n* = 3). Values between each pair of columns connected with a line marked with different stars on the top were significantly different (* *p* < 0.05, ** *p* < 0.01, *** *p* < 0.001). EPS-YW11: exopolysaccharide produced by *L. plantarum* YW11; CAS: casein.

**Table 1 foods-10-01153-t001:** The hardness, viscosity, cohesion, and adhesion of the EPS−YW11/CAS complex at EPS concentrations up to 1% (*w*/*v*).

EPS-YW11 (%)	Hardness (g)	Viscosity (mJ)	Cohesion	Adhesion (g)
0	71.3 ± 5.01a	0.03 ± 0.01d	0.64 ± 0.03a	42.6 ± 2.34a
0.25	66.3 ± 4.54a	0.09 ± 0.01bc	0.63 ± 0.02a	41.9 ± 2.25a
0.5	49.8 ± 3.58b	0.08 ± 0.01c	0.65 ± 0.01a	32.4 ± 2.08b
0.75	34.4 ± 2.64c	0.09 ± 0.02bc	0.67 ± 0.02a	23.2 ± 1.48c
1	29.0 ± 2.21d	0.11 ± 0.01a	0.66 ± 0.01a	19.2 ± 1.45c

The results are represented as mean ± SD (n = 3). Different letters in the same column indicate significant difference (*p* < 0.05).

## Data Availability

Not applicable.

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
