# Peer review of "Interaction of the Exopolysaccharide from Lactobacillus plantarum YW11 with Casein and Bioactivities of the Polymer Complex"

_foods, 2021, doi:10.3390/foods10061153_

Round 1

Reviewer 1 Report

In this manuscript, Zhang et al. evaluate the interaction of exopolysaccharides (EPS) from Lactobacillus plantarumYW11 and casein (CAS) during simulated fermentation process. They perform microrheological analysis, texture profile, water holding capacity, ζ-potential, particles sizes, antioxidant ability, bile binding capacity, anti-biofilm activity and morphology of EPS/CAS complex. The results presented in this study clearly showed that the addition of L. plantarumEPS on casein (3% w/v) increases the water holding capacity, viscoelasticity, particle size and ζ-potential, antibiofilm, antioxidation and bile acid binding capacity of mixture of casein and EPS product compared to CAS alone. I think this study will contribute to the dairy food research. 

Although issues covered in the manuscript are interesting, I have few major concerns:

  1. First, methods section seems incomplete. In several occasions the authors refer to previous studies, often authored by themselves, instead of providing a brief overview of what was done. It’s ok to refer to those publications for detailed information but brief outline enough to understand the methodology is necessary. Asking reader to go to a list of papers to figure out the approach is not fair.
  2. Second, data analysis section does not exist. The authors mention (line 183) that the analysis were performed in SPSS but never mention what exact analysis or a list of analyses were used. This section should provide what test (t-test, anova, regression, ….) was performed, what underlying assumptions were made to run that analysis, did the data meet those assumptions, and how the performance of analysis was evaluated. Results make very little sense in the absence of this section in methods.
  3. Another major issue I discovered is that the manuscript contains several paragraphs that are simply copied and pasted from previous publication by the same authors (reference # 15: Zhang, M.; Luo, T.; Zhao, X.; Hao, X.; Yang, Z. Interaction of exopolysaccharide produced by Lactobacillus plantarum YW11 with whey proteins and functionalities of the polymer complex. J. Food Sci. 2020, 85, 4141-4151). For example, you can look at the lines 57-59 of the manuscript. A few other paragraphs: 75-78, 89-94, 129-133 (there might be more but I stopped checking for them after this point). In my understanding copying and pasting from previous publications, including the authors own publications, is considered unethical and should be avoided. I would suggest the authors to carefully check their manuscript for such instances and rewrite the paragraphs.

I have a few specific comments:

  1. Line 12-13: Italicize the genus and species name - Lactobacillus plantarum.
  2. Please keep similar format of references in the text. See Line 125, 319, 322, 332, 367, 378…….
  3. Line 142 and 282: Please define FT-IR when mentioned for the first time in the manuscript.
  4. Line 335: Italicize genus and species names (do so in other places as well).
  5. Line 343: Change “exstence” to “existence”.
  6. Please provide a short description (or title) for each figure legend before describing each panel. In some figures, there were stars and lines on top of bars but never described what they represented. Anything shown in the figure should be defined in the legends. Also, provide the name of statistical test that was used to calculate significant differences between two or more groups. Finally, provide complete forms in the corresponding figure legends or notes under table for the abbreviations used in figures and table.
  7. Figure 2: Any significant effect of EPS concentration in WHC and others? Again, what was the statistical analysis?
  8. Figure 4: See above comments. Also, describe what those alphabets represent in the figure for Salmonella typhimurium. Generally, people use alphabets to identify significant differences between groups. Here, the authors seem to use asterisks to show such differences (though don’t define what those asterisks represent in legends). I don’t understand what those alphabets are doing once the differences are marked by asterisks.

Author Response

Dear Reviewer,

We have modified your suggestions and comments.Below are specific responses to each question.

In this manuscript, Zhang et al. evaluate the interaction of exopolysaccharides (EPS) from Lactobacillus plantarumYW11 and casein (CAS) during simulated fermentation process. They perform microrheological analysis, texture profile, water holding capacity, ζ-potential, particles sizes, antioxidant ability, bile binding capacity, anti-biofilm activity and morphology of EPS/CAS complex. The results presented in this study clearly showed that the addition of L. plantarum EPS on casein (3% w/v) increases the water holding capacity, viscoelasticity, particle size and ζ-potential, antibiofilm, antioxidation and bile acid binding capacity of mixture of casein and EPS product compared to CAS alone. I think this study will contribute to the dairy food research.

Although issues covered in the manuscript are interesting, I have few major concerns:

  1. First, methods section seems incomplete. In several occasions the authors refer to previous studies, often authored by themselves, instead of providing a brief overview of what was done. It’s ok to refer to those publications for detailed information but brief outline enough to understand the methodology is necessary. Asking reader to go to a list of papers to figure out the approach is not fair.

Response: Thank you very much for your suggestions. We have made a simple description of the method part, instead of just quoting other references directly.

  1. Second, data analysis section does not exist. The authors mention (line 183) that the analysis were performed in SPSS but never mention what exact analysis or a list of analyses were used. This section should provide what test (t-test, anova, regression, ….) was performed, what underlying assumptions were made to run that analysis, did the data meet those assumptions, and how the performance of analysis was evaluated. Results make very little sense in the absence of this section in methods.

Response: Thanks for your advice. More exact information of analyses has been added (Line 185-186). The experimental data were analyzed by one way ANOVA by using SPSS 22.0 software.

  1. Another major issue I discovered is that the manuscript contains several paragraphs that are simply copied and pasted from previous publication by the same authors (reference # 15:Zhang, M.; Luo, T.; Zhao, X.; Hao, X.; Yang, Z. Interaction of exopolysaccharide produced by Lactobacillus plantarum YW11 with whey proteins and functionalities of the polymer complex. J. Food Sci. 2020, 85, 4141-4151). For example, you can look at the lines 57-59 of the manuscript. A few other paragraphs: 75-78, 89-94, 129-133 (there might be more but I stopped checking for them after this point). In my understanding copying and pasting from previous publications, including the authors own publications, is considered unethical and should be avoided. I would suggest the authors to carefully check their manuscript for such instances and rewrite the paragraphs.

Response: Thanks for your advice. We have revised the whole manuscript so that no simple copying or pasting from what have been published earlier.

I have a few specific comments:

  1. Line 12-13: Italicize the genus and species name -Lactobacillus plantarum.

Response: Thanks a lot for the comments. The genus and species name -Lactobacillus plantarum has been italicized (line 12-13).

  1. Please keep similar format of references in the text. See Line 125, 319, 322, 332, 367, 378…….

Response: Thanks for your advice. The format of references in the text have been modified.

  1. Line 142 and 282: Please define FT-IR when mentioned for the first time in the manuscript.

Response: FI-IR has been defined with the complete name in the manuscript when mentioned for the first time (Line 142-143).

  1. Line 335: Italicize genus and species names (do so in other places as well).

Response: All genus and species names in the manuscript have been italicized.

  1. Line 343: Change “exstence” to “existence”.

Response: The spelling error has been corrected (Line 344).

  1. Please provide a short description (or title) for each figure legend before describing each panel. In some figures, there were stars and lines on top of bars but never described what they represented. Anything shown in the figure should be defined in the legends. Also, provide the name of statistical test that was used to calculate significant differences between two or more groups. Finally, provide complete forms in the corresponding figure legends or notes under table for the abbreviations used in figures and table.

Response: Thanks for the comments. The name of statistical test that was used to calculate significant differences between two or more groups has been added in the part of statistical analysis. The abbreviation used in the figure has been provided complete forms(Figure 2). The stars and lines on top of bars of figure 4 and figure 5 have been described in the figure notes.

  1. Figure 2: Any significant effect of EPS concentration in WHC and others? Again, what was the statistical analysis?

Response: The significant effects of EPS concentration in WHC, ζ-potential and particle size have been indicated in figure2 with different stars. The experimental data were analyzed by one way ANOVA by using SPSS 22.0 software.

  1. Figure 4: See above comments. Also, describe what those alphabets represent in the figure for Salmonella typhimurium. Generally, people use alphabets to identify significant differences between groups. Here, the authors seem to use asterisks to show such differences (though don’t define what those asterisks represent in legends). I don’t understand what those alphabets are doing once the differences are marked by asterisks.

Response: Those alphabets represented in the figure for Salmonella typhimurium have been deleted. The asterisks represented in legends have been defined in the figure notes.

Reviewer 2 Report

The paper was aimed at understanding the interactions and the potential bioactivities of the polymer complex formed by CAS and exopolysaccharides during a simulated fermentation process.

The paper address a very current and interest topic with an interesting approach. However, I have to express my doubts regarding some of the obtained results.

In fact, in the conclusion the authors state that this could find application in functional dairy products. In my opinion this is partially correct. In fact is well know that when fermentations are driven by microorganisms the coagulation kinetics are completely different, and also the position of cells themselves play an important role in the clot formation and in the conformation of casein agglomerates. Furthermore is also well know that the effect of the addition of EPS during a simulated fermentation, have a completely different impact, compared to the in situ synthesis of EPS from LAB.

As follows some comments on the text:

  • Line 35 (and throughoutthe text) delate the before EPS
  • Line 39.. due to the formation
  • Line 53 I suggest to add a very recent reference regarding the presence of EPS in fermented milk and viscosity, WHC etc.. Bancalari E, Alinovi M, Bottari B, Caligiani A, Mucchetti G and Gatti M (2020) Ability of a Wild Weissella Strain to Modify Viscosity of Fermented Milk. Microbiol. 10:3086. doi: 10.3389/fmicb.2019.03086
  • Line 54: I suggest to delate the entire sentence as it is an inappropriate example
  • Line 77-78 I suggest to modify the sentence as follows: … the strain was revitalized in SDM medium by two overnight sub-culturing at 37°C.
  • Line 77-78 Does the authors have performed a scaling up of fermentation to achieve the quantity of EPS necessary for the analysis? If yes, I suggest to add this part. Furthermore, the authors fail in providing the quantity of extracted EPS from the cultivation of plantarum.
  • Line 106: Can the author provide a better explanation about why the temperature of 42°C has been chosen for the analysis?
  • Line 147: … (TEM) analysis was…
  • Line 168 I think that this analysis are not in line with the scope of the paper. Despite could be an interesting information I think that the authors should focus on the real scope of the paper, furthermore it is also not well detailed. I suggest to remove this part from the paper
  • Line 203, when appear for the first time, abbreviations should be detailed
  • Line 223 Figure footnote: the abbreviations used should be better explained
  • Line 231: the authors state that “the viscosity… were not obviously affected”. Why obviously? In literature is extensively reported that the greater the amount of EPS, the highest the increase in viscosity. Could the authors explain why they state the contrary? I suggest some publications that you may add to the paper:
    • Ruas-Madiedo, P.,Tuinier, R., Kanning, M.,Zoon, P.(2002). Role of exopolysaccharides produced by Lactococcus lactis subsp. cremoris on the viscosity of fermented milks. https://doi.org/10.1016/S0958-6946(01)00161-3
    • De Vuyst, L., Zamfir, M., Mozzi,F., Adriany, T., Marshall, V., Degeest, B., Vaningelgem, F. (2003). Exopolysaccharide-producing Streptococcus thermophilus strains as functional starter cultures in the production of fermented milks. https://doi.org/10.1016/S0958-6946(03)00105-5
    • Bancalari, E., Alinovi, M., Bottari, B., Caligiani, A., Mucchetti, G., Gatti, M. (2020). Ability of a wild Weissella strain to modify viscosity of fermented milk. Microbiol.; Doi: https://doi.org/10.3389/fmicb.2019.03086
  • Line 234 I suggest to replace “curd” with clot
  • Line 241-243 the sentence is not well written, therefore is not clear. I suggest to rewrite to better clarify the notion.
  • Line 247: again I suggest to replace the term curd with clot
  • Line 328: again I suggest to remove this part
  • Line 368: …. Acids in a relation with the concentration.
  • Line 371: please remove the before EPS, and please check it throughout the text
  • Line 391: the same as 231

As a general comment I suggest to implement the result and discussion part with a more detailed description of the results respect to the recent  findings. In fact, despite a lot of analysis has been performed, it seems that the obtained results are poor.

Author Response

Dear Reviewer,

We have modified your suggestions and comments.Below are specific responses to each question.

The paper was aimed at understanding the interactions and the potential bioactivities of the polymer complex formed by CAS and exopolysaccharides during a simulated fermentation process.

The paper address a very current and interest topic with an interesting approach. However, I have to express my doubts regarding some of the obtained results.

In fact, in the conclusion the authors state that this could find application in functional dairy products. In my opinion this is partially correct. In fact is well know that when fermentations are driven by microorganisms the coagulation kinetics are completely different, and also the position of cells themselves play an important role in the clot formation and in the conformation of casein agglomerates. Furthermore is also well know that the effect of the addition of EPS during a simulated fermentation, have a completely different impact, compared to the in situ synthesis of EPS from LAB.

Response: Thank you very much for your questions. We agree that adding EPS directly is not the same as producing EPS directly during fermentation. Therefore, “... to further explore EPS-producing LAB..” has been modified as “...to further explore EPSs produced by LAB...” (line 411)

As follows some comments on the text:

  • Line 35 (and throughoutthe text) delatethe before EPS
  • Response: Thanks for the comments.“the” before EPS throughout text have been deleted.
  • Line 39.. due tothe formation
  • Response: “the” before formation has been added.
  • Line 53 I suggest to add a very recent reference regarding the presence of EPS in fermented milk and viscosity, WHC etc.. Bancalari E, Alinovi M, Bottari B, Caligiani A, Mucchetti G and Gatti M (2020) Ability of a Wild Weissella Strain to Modify Viscosity of Fermented Milk. Microbiol. 10:3086. doi: 10.3389/fmicb.2019.03086
  • Response: Thanks for your suggestion. The reference has been cited (Line 253-254).
  • Line 54: I suggest to delate the entire sentence as it is an inappropriate example
  • Response: Thanks a lot for the comments. The sentence has been delated.
  • Line 77-78 I suggest to modify the sentence as follows: … the strainwas revitalized in SDM medium by two overnight sub-culturing at 37°C.
  • Response: Thanks for your advice. The sentence has been modifiedas suggested.
  • Line 77-78 Does the authors have performed a scaling up of fermentation to achieve the quantity of EPS necessary for the analysis? If yes, I suggest to add this part. Furthermore, the authors fail in providing the quantity of extracted EPS from the cultivation of 
  • Response: The scaling-upfermentation was not performed in this study. The yield of the EPS at 131.26 mg/L produced in a semi-defined medium (SDM) shown in our previous study was mentioned in the revised manuscript and a reference was cited (line 58).
  • Line 106: Can the author provide a better explanation about why the temperature of 42°C has been chosen for the analysis?
  • Response: Thanks for the comments. The temperature of 42℃ is the common temperature for milk fermentationto produce yogurt, and there are corresponding references. Please refer to 20 and 21.
  • Line 147: … (TEM) analysiswas…
  • Response: Corrected to be(TEM) analysis” (Line 150).
  • Line 168 I think that this analysis are not in line with the scope of the paper. Despite could be an interesting information I think that the authors should focus on the real scope of the paper, furthermore it is also not well detailed. I suggest to remove this part from the paper
  • Response: Thanks for your suggestion. Since our previous study showed that EPS-YW11 possessed antioxidant activity, EPS-YW11/CAS was evaluated for antioxidant activity in this study. More detailed information of the method was added in the revised manuscript (line 170-174).
  • Line 203, when appear for the first time, abbreviations should be detailed
  • Response:The detailed names of the abbreviation has been added. (Line 204-205)
  • Line 223 Figure footnote: the abbreviations used should be better explained.
  • Response: The detailed namesof the abbreviations have been added (Line 204-205) and in figure footnote.
  • Line 231: the authors state that “the viscosity… were not obviously affected”. Why obviously? In literature is extensively reported that the greater the amount of EPS, the highest the increase in viscosity. Could the authors explain why they state the contrary? I suggest some publications that you may add to the paper:
    • Ruas-Madiedo, P.,Tuinier, R., Kanning, M.,Zoon, P.(2002). Role of exopolysaccharides produced by Lactococcus lactis subsp. cremoris on the viscosity of fermented milks. https://doi.org/10.1016/S0958-6946(01)00161-3
    • De Vuyst, L., Zamfir, M., Mozzi,F., Adriany, T., Marshall, V., Degeest, B., Vaningelgem, F. (2003). Exopolysaccharide-producing Streptococcus thermophilus strains as functional starter cultures in the production of fermented milks. https://doi.org/10.1016/S0958-6946(03)00105-5
    • Bancalari, E., Alinovi, M., Bottari, B., Caligiani, A., Mucchetti, G., Gatti, M. (2020). Ability of a wild Weissellastrain to modify viscosity of fermented milk. Microbiol.; Doi: https://doi.org/10.3389/fmicb.2019.03086

Response: Thanks a lot for the comments. These references have been added in the revised manuscript. We have carried out statistical analysis on this part of data and confirmed the increased viscosity with greater amount of EPS, and this has been revised it in the revised manuscript.

  • Line 234 I suggest to replace “curd” with clot
  • Response: Thanks for the suggestion. All “curd” in this article have been replaced with “clot”.
  • Line 241-243 the sentence is not well written, therefore is not clear. I suggest to rewrite to better clarify the notion.
  • Response:We've replaced “The better the WHC of the gel, the stronger the force of the gel to bind water, and the better the stability of the gel”. (line 244-245)
  • Line 247: again I suggest to replace the term curd with clot
  • Response: Thanks for the suggestion. All “curd” in this article have been replaced with “clot”.
  • Line 328: again I suggest to remove this part
  • Response: Thanks for the suggestons. Part of the text has been deleted.
  • Line 368: …. Acids in a relation with the concentration.
  • ResponseWe've completedthis sentence.(369-370)
  • Line 371: please remove the before EPS, and please check it throughout the text
  • Response: Thanks for your advice. “the” before EPS has been removedthroughout the tex.
  • Line 391: the same as 231
  • Response:We have revised this part of the conclusion.

As a general comment I suggest to implement the result and discussion part with a more detailed description of the results respect to the recent findings. In fact, despite a lot of analysis has been performed, it seems that the obtained results are poor.

Response: Thanks a lot for the comments. More detailed description was performed in the revised manuscript.

Reviewer 3 Report

This study covers the interaction between EPS and casein. Overall this was well written and the bioactivities were well interpreted based on the results. But, there is a little doubt on the anti-biofilm assay whether it was evaluated the inhibition on the planktonic-to-biofilm transit or biofilm-formed cells. It is not much clear how to measure the ability of bacteria to form biofilm.

Author Response

Dear Reviewer,

We have modified your suggestions and comments.Below are specific responses to each question.

This study covers the interaction between EPS and casein. Overall this was well written and the bioactivities were well interpreted based on the results. But, there is a little doubt on the anti-biofilm assay whether it was evaluated the inhibition on the planktonic-to-biofilm transit or biofilm-formed cells. It is not much clear how to measure the ability of bacteria to form biofilm.

Response: Thank you very much for your comments. In this study, inhibition on the planktonic-to-biofilm transit was not performed. Regarding the biofilm-formed cells, formation of the biofilm on the bacterial cell surface was evaluated in this study. More detailed procedure of the anti-biofilm assay was added to clarify the assay in the revised manuscript (line 182-185).

Round 2

Reviewer 1 Report

I still data analysis section is incomplete. Is there any post hoc comparisons?

Author Response

Thanks a lot for your reminding. The Fisher’s least significant difference (LSD) method was used for post hoc comparisons. This has been completed in the Statistical analysis part.(Line 192)

Reviewer 2 Report

The authors provided the required modifications to the text.

Author Response

Thank the reviewers for these precious comments concerning my manuscript entitled “Interaction of the exopolysaccharide from Lactobacillus plantarum YW11 with casein and bioactivities of the polymer complex”. These comments are all valuable and very helpful for revising and improving my paper, as well as the important guiding significance to our researches.